# A Biomimetic Treadmill-Driven Ankle Exoskeleton: A Study in Able-Bodied Individuals

**DOI:** 10.3390/biomimetics10090635

**Published:** 2025-09-21

**Authors:** Matej Tomc, Matjaž Zadravec, Andrej Olenšek, Zlatko Matjačić

**Affiliations:** 1University Rehabilitation Institute Republic of Slovenia-Soča, 1000 Ljubljana, Slovenia; matjaz.zadravec@ir-rs.si (M.Z.); andrej.olensek@ir-rs.si (A.O.); zlatko.matjacic@ir-rs.si (Z.M.); 2Faculty of Electrical Engineering, University of Ljubljana, 1000 Ljubljana, Slovenia; 3Department of Biomechanical Engineering, Faculty of Engineering Technology, University of Twente, 7500 AE Enschede, The Netherlands

**Keywords:** rehabilitation, gait, wearable robotics, ankle exoskeletons, lower-limb joints, biomechanics, push-off, plantarflexors

## Abstract

Despite rapid growth in the body of research on ankle exoskeletons, we have so far not seen their massive adoption in clinical rehabilitation. We foresee that an ankle exo best suited to rehabilitation use should possess the power generation capabilities of state-of-the-art active exos as well as the simplistic control and inherently suitable assistance timing seen in passive exos. In this paper we present and evaluate our attempt to create such a hybrid device: an Ankle Exoskeleton with Treadmill Actuation for Push-off Assistance. Using our device, we assisted a group of able-bodied individuals in generating ankle plantarflexion torque and power while measuring changes in biomechanics and electromyographic activity. Changes were mostly contained to the ankle joint, where a reduction in biological power and torque generation was observed in proportion to provided exo assistance. Assistance was comparable to state-of-the-art active exos in both timing and torque trajectory shape and well synchronized with the user’s own biological efforts, despite using a very simplistic controller.

## 1. Introduction

### 1.1. Background

In recent years, we have seen rapid growth in the field of wearable robotic devices that assist in locomotion. With the largest share of positive mechanical power for forward center of mass propulsion during gait being generated at the ankle joint [1,2], many of these devices were made in the form of ankle exos.

The first to break the metabolic cost barrier, enabling more energetically efficient walking compared to without an exo, was Malcolm et al. in 2013 [3]. The barrier was soon after also broken with a fully passive ankle exo [4]. Since then, numerous studies have reported such an achievement, becoming more and more successful in their pursuit of metabolic energy savings [5].

But while many of these studies state the goal of future use in rehabilitation, the increasingly complex mechanisms and algorithms used suggest that the field has steadily moved towards able-bodied human augmentation while neglecting practical limitations inherent to transfers to clinical environments and patient populations. Indeed, while the research field has grown expansive in recent years, we have not yet seen a breakthrough into daily clinical practice or the impressive results in the field of rehabilitation that researchers have hoped for.

It is clear that the control aspect of ankle exos is the key for symbiotic functioning with a human. A state-of-the-art example of a control approach that has shown great promise recently has been human-in-the-loop optimization (HILO), optimizing for either an individual’s metabolic cost [6,7,8] or chosen muscle activity metrics [9,10]. While the approach resulted in very performant controllers, the optimization process was often computationally power-demanding and time-consuming. Various other factors were not considered, such as users’ comfort and, in the case of neurologically impaired subjects, high step-to-step variability in walking patterns, which may further complicate the optimization process. Further development of HILO-based control approaches foresees utilization of high-scale personalized musculoskeletal modeling that would allow for offline training of HILO engines through simulations using real-time walking data. While more efficient in terms of patient time, it indicates even further increasing demands on technology, which in turn may drive modern exos further away from practical use. Even if we assume that the described approach may in the future become feasible for individualized use in able-bodied subjects, it seems that great variability surrounding the gait of various neurologically impaired populations would preclude successful implementation in clinical applications.

On the opposite end of the research field, there are fully passive ankle exos (e.g., [4,11]), which have also been shown to reduce the metabolic cost of walking in able-bodied users while not requiring any control algorithm at all. Utilizing biomimetic designs, passive exos can inherently apply appropriately shaped and well-timed assistive torque patterns in synchrony with the user’s own efforts, which is otherwise a crucial and difficult-to-solve control problem in active exos [12,13]. Passive exos are generally also lighter in weight and less complex compared to active ones. The main limitation of passive ankle exos is that they can only redistribute power generation among body segments and gait phases but cannot inject any additional externally sourced energy into gait, which greatly limits their practical potential, especially in rehabilitation.

### 1.2. Study Aim

In our estimations, an exo aimed at use in rehabilitation should ideally possess the lightweight and inherently synchronous assistance features of passive exos while also meeting the additional power requirements of patients, especially during push-off, which can usually be supplied by active ankle exos. Preferably, the design should also address practical concerns that inhibit the adoption of modern exos in clinical practice, such as cost, mechanical and control complexity. Our attempts to combine these features are materialized in a novel device called the Ankle Exoskeleton with Treadmill Actuation for Push-off Assistance (AN-EXTRA-Push).

In this study, we evaluated the performance of AN-EXTRA-Push on a group of able-bodied individuals. We hypothesized that AN-EXTRA-Push should have the power-generating capabilities typical for a state-of-the-art active ankle exo, while retaining the inherently synchronizing features of passive exos. To properly assess the synchrony between exo assistance and the user’s own biological torque/power, we formulated a new objective metric for assistance synchrony.

## 2. Materials and Methods

### 2.1. AN-EXTRA-Push

AN-EXTRA-Push is a biomimetically inspired ankle exo intended for gait rehabilitation for people with neuromuscular impairments (e.g., post stroke), made to be used on a treadmill in a clinical environment. The concept for the device was first introduced in [14], where we tested its feasibility in silico. It was later further iteratively developed in [15,16,17]. Here, we give a brief overview of the latest version of our exo.

AN-EXTRA-Push consists of an orthosis (parts A, B, and C in Figure 1) worn by the user, a braking and pretensioning mechanism (parts F and G in Figure 1) located in front of the treadmill (H), and an elastic tendon (D and E in Figure 1) connecting the two. The orthosis has one degree of freedom. Its rotary joint is aligned as best as possible with the biological ankle joint sagittal plane rotation axis.

On one end, the elastic tendon (D) is routed through an electromechanical brake (F) and connected to a wall with an additional constant force spring (G) in series that prevents the elastic tendon from going slack. The other end of the tendon is split in two and wound on the outer pair of a trio of concentric spools (C) at the rear end of the orthosis. The three spools form a rigid structure. They rotate around their common axis. A rigid cable is wound around the middle spool and connected to the shank part of the orthosis (A). A pulling force in the elastic tendon (D) is transferred to the rigid cable (E) via the concentric spools (C), causing the shank (A) and foot (B) parts of the orthosis to be pulled together, creating torque at the rotary joint. Spools in C have different radii, defining the force transmission ratio, explained in detail in [16].

The total weight of the parts of the device that were worn by the user was 1.45 kg. For users weighing over 72.5 kg, this does not exceed the 2% body weight threshold given in the literature [18,19], which marks the point where the gait pattern is significantly affected by the added weight at the ankle.

The device was designed to imitate the combined function of the soleus muscle and the Achilles tendon [16]. The normal AN-EXTRA-Push operation is summarized in a flowchart in Figure 1. Two representative moments (T1 and T2) during the gait cycle are depicted in the top left portion of this figure. During the swing phase, the brake is disengaged, and only a compliant constant force spring pulls on the tendon to keep it taut. During stance, we engage the brake. Posterior movement of the leg by the treadmill combined with ankle dorsiflexion stretches the elastic tendon, storing elastic energy, building up force, and consequently building plantarflexion torque. Peak force in the tendon occurs right before push-off. After push-off initiation, the ankle plantarflexes rapidly, rapidly releasing the stored energy in a burst of power. The brake is then disengaged, making it ready for the next cycle. Details about when the brake engagement/disengagement should occur were explored in silico in [14] and confirmed in preliminary experiments.

### 2.2. Study Participants

Twelve healthy adults (all male, age: 27.3 ± 0.9 years, height: 178.1 ± 5.1 cm, weight: 80.7 ± 7.9 kg (mean ± SD), 7 wore EU size 42 shoes, 5 wore EU size 44 shoes, all naïve subjects) participated in this study. All participants provided informed consent. This study was approved by the National Medical Ethics Committee of the Republic of Slovenia.

A full table of participant demographics is available in Appendix A (Appendix A).

Due to a mechanical malfunction of the device, measurements recorded for Subject 2 were corrupted and thus excluded from the results.

### 2.3. Study Protocol

Participants wore AN-EXTRA-Push on their left leg. They walked under 11 different experimental conditions: (1) ZeroTorque—wearing only the additional weight of the exo while untethered, (2) Transparent—system pre-tensioned never engaged (exo connected to constant spring via inelastic rope), and (3–11) Active assistance conditions comprised of all combinations of three brake engagement timings (Early = 20.0% GC, Middle = 25.7% GC, Late = 31.4% GC) and three elastic tendon stiffnesses (k1 = 33 N/m, k2 = 67 N/m, k3 = 105 N/m), e.g., Middle_k3. The stiffness levels were the same for all participants and were not scaled to their body weight. During preliminary testing, walking without the exo was compared to the ZeroTorque condition. Only minor differences at the knee joint were present, which we could best explain just by the added weight at the ankle. An experimental condition of walking without the exo was therefore not tested in this study to avoid the additional donning/doffing time and differences in added weight between conditions.

Throughout the entire experiment, treadmill speed was set to 0.8 m/s. While substantially lower than the normal preferred walking speed [20], it was chosen to allow for closer future comparisons to gaits of people with neuromuscular impairments (e.g., stroke [21]).

The participants familiarized themselves with the setup, first by walking on a treadmill for 10 min without the exo, then 4 min in the ZeroTorque condition, 2 min in the Transparent condition, and finally 4 min with active assistance in the Middle_k2 condition (Figure 2a).

Familiarization was followed immediately by measurements. Experimental conditions were distributed in a pseudo-random order (Figure 2c). The ZeroTorque condition was always measured first and repeated again as the last condition. Other experimental conditions were randomly interspersed with conditions that feature the same tendon stiffness always grouped together to minimize interruptions, since the stiffness was adjusted with a manual intervention. Each experimental condition lasted 2 min, followed by 30 s of walking with the brake off (equal to the Transparent condition) to allow the participants to return to their usual gait pattern before continuing.

### 2.4. Biomechanics and EMG Measurements

Sixteen reflective markers were placed onto the participants’ bodies in accordance with the Plug-In Gait lower body protocol [22]. Their position was tracked at 120 Hz using the Optitrack six-camera motion capture system (NaturalPoint, Inc., Corvallis, OR, USA). The raw data were first filtered using a 10 Hz low-pass filter in Motive (NaturalPoint, Inc., Corvallis, OR, USA) and then merged with the subjects’ anthropometric data in a custom Python 3.11 script to calculate direct kinematics.

During this study, subjects walked on a custom-built split-belt treadmill. The left belt, on which they walked with their leg wearing the exo, was instrumented using four triaxial force transducers (K3D120, ME Systeme GmbH, Hennigsdorf, Germany). Ground reaction force and center of pressure data were measured at 250 Hz. The right belt served as an auxiliary walking surface and was not instrumented. The custom treadmill setup carried certain limitations, further discussed in the Discussion section.

AN-EXTRA-Push was equipped with a U9C force sensor (HBM, Darmstadt, Germany). It measured the force in the rigid cable at the back of the orthosis, which equals five times the force in the elastic tendon. The force measurement was performed at 250 Hz. We used a Beckhoff CX5020 industrial computer (Beckhoff Automation, Verl, Germany) to record the force data from the treadmill and the back of AN-EXTRA-Push.

Muscle activity of m. soleus, m. gastrocnemius lateralis, m. tibialis anterior, m. vastus medialis, m. biceps femoris, and m. rectus femoris of both legs was measured using surface electromyography (EMG) (TelemyoMini 16, Neurodata, Vienna, Austria). We used 3M Red Dot 2560 (3M, Saint Paul, MN, USA) electrodes (interelectrode distance: 4 mm). EMG data were recorded at 1 kHz. An example of electrode and reflective marker placements can be seen in Figure 2b.

### 2.5. Data Processing

The Newton–Euler inverse dynamics algorithm was used to calculate torques and powers at the ankle, knee, and hip of the left leg. The use of AN-EXTRA-Push was accounted for by recalculating the mass and inertia matrix of the shank and foot segments. The force measurement in the rigid cable at the back of the orthosis was used to calculate the additional external torque in the sagittal plane at the ankle joint and the external force of the elastic tendon pulling on the orthosis in the direction of the brake. The contributions of AN-EXTRA-Push towards the total kinetics were then subtracted from the total to calculate the biological contributions. Spatiotemporal parameters (e.g., stride length, stance phase time, etc.) were also calculated for every measurement.

Muscle excitation signals were calculated from raw EMG signals by first manually removing the movement artifacts and outliers, then filtering with a bandpass (20–140 Hz) and notch (49–51 Hz) filter, full-wave rectifying, and finally calculating the envelopes using moving average (window width: 150 ms). For each participant, signals were then normalized to the peak value of the average stride in the ZeroTorque condition for each muscle.

All data processing was performed in Python. We synced up the kinematics, kinetics, and muscle excitations, then segmented the data into gait cycles. Measurements from the ZeroTorque condition, which was the only condition we repeated, were merged into a single file so that the steps from each measurement instance alternated.

Only 20 strides (from the 20th to 39th) from each walking bout were used for further analysis to ensure only steady-state walking was considered.

### 2.6. Statistics

Biomechanically interesting scalar features were extracted from timeseries data for joint angles, torques, powers, and muscle excitations. Those included peak and trough values of time series, as well as signal integrals. When applicable, only parts of the gait cycle were used for feature extraction: 0 to 55% of GC is labeled stance phase, and 30 to 50% of GC is labeled midstance, even though the actual phase transitions vary stride-to-stride and subject-to-subject. Similarly, only positive or negative torque and power values are integrated where applicable (e.g., ankle power, where increases in both positive and negative power might cancel out over the GC).

To objectively report the synchrony between ankle exo assistance and the user’s own biological efforts, we defined a new metric. We calculated the time lag between the peak biological ankle torque and power and peak exo-assistive torque and power, respectively. For well-synchronized exo assistance and healthy-like shapes of the biological torque and power trajectories, the time lag should be close to zero. This metric does not consider the shape of the assistive torque and power; however, it still gives a quick and simple way to compare the success of synchronizing the exo to the biological musculotendon system. Despite the well-known importance of synchrony, we have not encountered any such measure in the literature.

We compared the characteristic scalar features across all eleven experimental conditions. We assessed differences using a one-way repeated measures ANOVA, with a significance threshold of α = 0.05.

To evaluate the sphericity assumption, we performed Mauchly’s test. If sphericity was violated, we applied the Greenhouse–Geisser correction to adjust the degrees of freedom and preserve the validity of the ANOVA results.

Whenever the ANOVA indicated a significant effect of condition, we conducted pairwise t-tests for all condition pairs. To mitigate the risk of false positives from multiple comparisons, we controlled the false discovery rate by adjusting *p*-values with a Benjamini–Hochberg correction.

Our highly anthropometrically homogenous study cohort is the consequence of shoe size limitations of AN-EXTRA-Push, which we address in detail in the limitations subsection of the Discussion. Additionally, the *Intersubject Variability* Subsection was added to the Results Section, where differences between the subjects’ gaits are objectively evaluated.

In the next section, only a selected portion of statistical results are presented. All the RM ANOVA and post hoc test results are compiled in the Appendix A.

When summarizing the results of post hoc tests in this text, we sometimes group the conditions with respect to their common stiffness level (k1, k2, or k3) or brake engagement timing (later abbreviated as “timing”; Early, Middle, or Late). References to groups are established for the sake of brevity of reporting—all conditions are treated as separate during statistical analysis.

## 3. Results

### 3.1. Ankle Biomechanics

AN-EXTRA-Push assistance slightly altered ankle kinematics (Figure 3A). Peak dorsiflexion was reduced by up to 3° compared to the ZeroTorque condition. Changes were statistically significant (F(3.10, 31.02) = 13.48, *p* = 7.30 × 10^−8^, partial η^2^ = 0.574) under different stiffness levels, but not timings.

Figure 3B,C show the biological and exo contributions to ankle torque. Depending on the experimental condition, AN-EXTRA-Push assistance contributed from 10.3% (Late_k1) to 28.5% (Early_k3) of total peak ankle torque. With different timings and stiffness levels, varied and significantly different levels of assistive torque were provided by the exo (F(10, 100) = 233.03, *p* = 1.4 × 10^−64^, partial η^2^ = 0.959 integrated over GC and F(10, 100) = 230.86, *p* = 2.19 × 10^−64^, partial η^2^ = 0.958 for peaks), as seen in Figure 3D (average value calculated as integral divided by duration) and Figure 3E. All post hoc pairwise tests also show significant differences (all *p* < 0.001). Across all experiments, higher stiffness values (k3 > k2 > k1) and earlier timings (Early > Middle > Late) resulted in higher peak exo torque values—and therefore higher levels of assistance—with the system being much more sensitive to the investigated stiffness changes compared to timing changes.

Increased exo assistance resulted in decreased biological contributions without an increase in total ankle torque. The total sum of biological and exo contributions remained largely unchanged (only the ZeroTorque condition marginally differed from others (all *p* > 0.02)). Decreased biological torque was observed over the entire GC (integral over GC: F(10, 100) = 13.48, *p* = 1.07 × 10^−14^, partial η^2^ = 0.574) (Figure 3D) as well as at the peaks: F(10, 100) = 60.31, *p* = 7.75 × 10^−38^, partial η^2^ = 0.858) (Figure 3E). Post hoc tests revealed that increases in stiffness levels contributed more significantly to biological torque reduction (all *p* < 0.005) compared to different brake engagement timings (all *p* < 0.05).

The additional plantarflexion torque from the exo and slightly altered kinematics resulted in increased positive (integral over GC: F(10, 100) = 186.51, *p* = 5.52 × 10^−60^, partial η^2^ = 0.949) and negative (integral over GC: F(10, 100) = 96.00, *p* = 1.16 × 10^−46^, partial η^2^ = 0.906) exo ankle power contributions (Figure 3I), proportional to differences in assistance levels between conditions. Biological ankle power decreased (Figure 3F)—both positive (integral over GC: F(10, 100) = 42.64, *p* = 1.19 × 10^−31^, partial η^2^ = 0.810) and negative (integral over GC: F(10, 100) = 48.69, *p* = 5.68 × 10^−34^, partial η^2^ = 0.830) (Figure 3H). The largest reduction in positive biological ankle power was observed during Early_k3. During Early_k3, 35.6% of total positive ankle power was provided by the exo. Compared to ZeroTorque, the integral of positive biological ankle power was reduced by 43.5%. Total ankle power (sum of biological and exo contributions) was slightly reduced with increased stiffness, both in the positive and negative directions, and was largely unaffected by timings.

### 3.2. Knee Biomechanics

With increased exo assistance, knee kinematics (Figure 4A) shifted slightly towards increased peak extension during midstance (up to about 3°, F(10, 100) = 11.60, *p* = 5.32 × 10^−13^, partial η^2^ = 0.537) and towards increased peak flexion in swing (up to about 3°, although without clear correlation with stiffness or timing changes).

With increased exo assistance, knee torque (Figure 4B) during midstance shifted from extension towards flexion (midstance trough: F(10, 100) = 12.73, *p* = 4.86 × 10^−14^, partial η^2^ = 0.560, post hoc tests show significance for most condition comparisons). With the relatively low angular velocity at the knee joint during midstance, torque differences did not translate into substantial power differences. RM-ANOVA shows significance in the integral of negative knee power (Figure 4C) over GC (F(10, 100) = 6.73, *p* = 6.25 × 10^−8^, partial η^2^ = 0.402), but post hoc tests reveal that this is mostly due to the Transparent condition being an outlier.

### 3.3. Hip Biomechanics

The hip joint appeared largely unaffected by the presence of AN-EXTRA-Push. Hip joint angle (Figure 5A) and torque (Figure 5B) trajectories remained stable across all conditions. RM ANOVA on hip torque integral reached statistical significance (F(10, 100) = 3.33, *p* = 9.02 × 10^−4^, partial η^2^ = 0.250), but only due to marginal differences between ZeroTorque and other conditions (post hoc *p*-values of around 0.01). A weak shift towards more positive hip power (Figure 5C) was observed during stance and push-off (hip power integral: F(10, 100) = 4.34, *p* < 0.0001, partial η^2^ = 0.303) but without clear correlation with increased exo assistance.

### 3.4. Electromyography

Muscle excitation timeseries are shown in Figure 6 for the soleus, gastrocnemius, and tibialis anterior of the left leg (exo-equipped leg). The rest are shown in Appendix A in Appendix A.

Soleus excitation was reduced, especially during push-off (the peak of the excitation signal precedes the peak of positive ankle power due to the inherent electromechanical delay). Both the integral over the GC (F(10, 100) = 13.63, *p* = 0.000008, partial η^2^ = 0.577) and peak value (F(10, 100) = 14.96, *p* < 0.000001, partial η^2^ = 0.599) RM-ANOVA analyses confirm statistical differences between conditions, with post hoc tests showing statistical significance mostly between different stiffness conditions.

The dependence of the decrease in gastrocnemius muscle activity on exo assistance during push-off was less clear (integral over GC: F(2.44, 24.45) = 4.72, *p* = 1.38 × 10^−2^, partial η^2^ = 0.320, peak: F(10, 100) = 4.64, *p* = 2.00 × 10^−5^, partial η2 = 0.317). Post hoc tests mostly indicate differences between ZeroTorque and other conditions (*p* between 0.0021 and 0.05), but not between active assistance conditions themselves.

Muscle excitation of tibialis anterior remained largely unchanged across experimental conditions. Figure 6C indicates a marginal increase during stance and after push-off, but the overall changes in integral over GC did not reach statistical significance.

Neither peaks nor integrals over GC differed significantly across conditions for the other measured muscles, including the contralateral (right) leg. The only exception was the biceps femoris of the right leg with marginal statistical differences (*p* > 0.03) between some conditions (see Appendix A in Appendix A).

### 3.5. Contralateral (Right) Leg

Across all active conditions, the kinematics and muscle excitations of the contralateral (right) leg remained substantially similar to those in the ZeroTorque condition. Results for the contralateral leg are shown in Appendix A in Appendix A.

### 3.6. Intersubject Variability

Across all subjects and experimental conditions, the average gait cycle duration was 1.41 ± 0.06 s (mean ± SD), of which the stance phase took 0.89 ± 0.04 s (63.2 ± 3.7% GC).

Clear differences were present between subjects in gait kinematics and kinetics, revealing large subject-to-subject gait pattern diversity. For all subjects, narrow standard deviation bands indicate that data were recorded during steady-state gait with little step-to-step variability.

All graphs of recorded joint kinematics and kinetics for individual participants during the ZeroTorque and Early_k3 conditions are shown in Appendix A.

### 3.7. Assistance Synchrony

Histograms in Figure 7 and Figure 8 show the synchrony between the exo and the user’s own musculotendon system, using our newly defined metric of assistance synchrony (i.e., time lag between the peak exo’s assistive and the user’s biological push-off torques/powers).

An AN-EXTRA-Push torque peak occurred on average 0.96 ± 0.07% GC (mean ± standard error of mean) before the biological torque peak (Figure 7). Peak exo power reached on average 2.41 ± 0.04% GC after the biological peak (Figure 8). These results represent the average synchrony across all eleven valid subjects and nine active assistance conditions.

Torque and power synchrony distribution are depicted twice in Figure 7 and Figure 8. On the left, the histograms are colored by grouping data from the same subject across all conditions, and vice versa on the right. While some statistical differences can be observed between synchrony distributions of different subjects or conditions, none of them are outliers and are functionally well summarized by the total mean and standard error of the mean.

## 4. Discussion

The key result validating that the AN-EXTRA-Push was performing its primary intended function is found in Figure 3, showing that the exo was able to provide an adjustable level of ankle plantarflexion assistance based on the choices of brake engagement timing and elastic tendon stiffness values. Depending on the experimental condition, exo torque peaks ranged from 10.3% to 28.5% of total ankle plantarflexion torque peaks at the extremes, with the other condition spaced in between. In a clinical use case, the therapist could utilize this range to set the appropriate assistance level of training to follow the assist-as-needed paradigm in rehabilitation. The tested tendon stiffness values had a larger effect on the assistance level of the exo compared to the brake engagement timings, which can be consistently observed across most tests in our statistical analysis.

With the increased exo contribution, the biological contribution was reduced in equal measure, leading to a largely unchanged total ankle torque. The peak biological ankle plantarflexion torque reduction due to external assistance in our study was 28.5%. For comparison, the peak biological ankle plantarflexion torque reduction observed in a study featuring healthy participants using an exo with a human-in-the-loop optimized controller was 31% [7]. This reduction was achieved after multiple days of individualized controller HILO and user adaptation [23]. Further increases in assistive torque levels did not lead to additional metabolic cost savings, which was the primary optimization criterion. With a generic assistive torque trajectory, developed in a pilot study leading up to studies [7,23], the peak biological torque reduction was 28%. Similar peak values were reported in other literature primarily focused on human augmentation ([24,25,26,27], in one case even reaching 47% [28], but that demanded significant changes in joint kinematics).

Among the devices focused on rehabilitation use cases, reported assistive torque values are lower: in a study by Orekhov et al. [29], only 0.03 Nm/kg of assistive torque provided with an active ankle exo to patients with cerebral palsy was enough to promote higher gait speed and lower the soleus muscle activation. The device used was battery powered, weighing 2.07 kg, with the majority of the weight accounted for by the battery pack strapped to the user’s waist.

An example of an exo aimed at stroke rehabilitation can be found in [30]. Awad et al. created a soft robotic exosuit capable of assisting with ankle plantarflexion and dorsiflexion torque. The battery-powered Bowden cable-based device weighed 3.1 kg. The majority of the weight was worn at the waist, with about 0.9 kg distributed down the leg. Using this exosuit, stroke survivors walked with improved symmetry, lower metabolic energy consumption, and a more healthy-like gait pattern. These results were achieved at only 0,15 Nm/kg (i.e., 12% total) assistive plantarflexion ankle torque.

In this study, AN-EXTRA-Push users modified their gait pattern to accommodate for the added exo plantarflexion torque. Changes in kinematics that we observed closely resemble those reported in the literature, where they were put under similar external torque conditions. During the stance phase, increased plantarflexion torque generally resulted in decreased ankle dorsiflexion angle [31,32,33,34]. After push-off, ankle plantarflexion angle was generally increased [31,32,33,35]. At other joints, kinematics changes were minor, if at all noticeable [7,24,27,31,32,33,34,35].

The effects of AN-EXTRA-Push assistance were also almost entirely isolated, as intended by the design, to the ankle joint of the leg equipped with the exo. No changes were observed at the hip. Study participants walked with slightly more extended knees on both legs during the stance phase. On the side equipped with the exo, additional knee flexion during early parts of the swing was also observed, likely to compensate for the slight shift in ankle angle towards plantarflexion and still ensure sufficient toe clearance.

The move towards increased knee extension is likely explained by the well-known ankle plantarflexion–knee extension coupling [36,37]. During stance, the external plantarflexion torque at the ankle results in tibia rotation since the foot is constrained by the treadmill surface and cannot move. The change is likely mirrored on the contralateral side by the users to maintain symmetry.

Ankle exos from the literature usually initiate assistance later on in the stance phase and therefore do not encounter this phenomenon. A previously mentioned rare example of an active exo by Awad et al., where earlier assistance starts were investigated, can be found in [30]. For some stroke survivors that participated in that study, earlier assistance helped improve their gait pattern, while for others it had the opposite effect.

With the added external plantarflexion torque, AN-EXTRA-Push provided additional power to the ankle joint—negative during early and midstance and positive during push-off. The total power integrated over the gait cycle remained largely unchanged. Study participants utilized the additional external power of the exo to reduce their own biological power generation. These observations are consistent with the literature [4,7,27,31,32,33,34].

Observed changes in ankle biomechanics are also well explained by our EMG measurement results. Soleus muscle excitation was reduced in proportion to the AN-EXTRA-Push assistance level. Similar results are consistently reported across studies from literature, where external plantarflexion torque was applied to healthy humans during gait [4,7,24,27,31,33,35]. From the point of view of designing a device for training in rehabilitation, the hope is that the externally provided additional plantarflexion torque will, in cases of users with neuromuscular deficits, supplement the lack in plantarflexion torque generation abilities while promoting optimal patient effort through a well-selected assistance level, following the principle of the assist-as-needed paradigm.

Just like the AN-EXTRA-Push mechanism, the soleus muscle and the Achilles tendon span the ankle joint, and the forces that are exerted by the muscle result in plantarflexion torque. The medial and lateral heads of the gastrocnemius muscle also connect to the Achilles tendon and, together with the soleus muscle, share the vast majority of the burden of biological plantarflexion torque generation. In contrast to soleus and AN-EXTRA-Push, gastrocnemius spans the knee as well as the ankle joint. It is therefore not surprising that the correlation between EMG measurements and AN-EXTRA-Push assistance levels holds firmer for the soleus compared to the gastrocnemius muscle. In similar fashion to our reasoning, in the literature, most studies with plantarflexion torque exo-assistance report reduced gastrocnemius activity [7,24,31,33,35], but not all do [4,27].

In the same literature, increased tibialis anterior (the main dorsiflexor and antagonist to soleus) muscle activity is also often reported [4,7,27,35,38]. This can in most cases be attributed to co-contraction of the plantarflexor-dorsiflexor agonist–antagonist pair. Co-contractions increase the ankle joint stiffness and are likely present due to the exo user seeking additional stability. In HILO studies focused on metabolic energy consumption minimization, increases in co-contraction were observed with higher assistance levels [7] but would generally lessen with user training [31]. Even at maximum assistance levels, no co-contractions have been observed in our study. The only relevant increase in tibialis anterior activity connected to AN-EXTRA-Push assistance coincides with the swing phase when the foot was returned from the overly plantarflexed position after push-off.

The importance of proper assistance timing is often discussed in the literature but is not formally or explicitly defined [3,12,13,39,40]. In recent HILO studies, the timing of, e.g., the start, the peak, or both of the assistance torque trajectory was varied until a torque trajectory with minimal metabolic cost was found [6,7,8]. Similar optimization techniques were also attempted to minimize chosen muscle activity metrics [9,10]. In both cases, once the optimization was completed for the individual, the chosen control law (torque trajectory deemed optimal) was applied to the exo for the rest of the experiment.

The “control law” of AN-EXTRA-Push is baked into its mechanical design, so the peak timing of the torque trajectory cannot be optimized for, only evaluated. Our newly proposed metric of assistance synchrony is helpful in that regard but does not allow for direct comparisons with studies from literature, since it hasn’t been reported. We attempted to calculate the assistance synchrony metric for a few relevant example studies in the field based on publicly accessible experimental data.

In [7], which features HILO-minimizing metabolic energy cost and uses an active ankle exo, the peak exo torque reached 9.5% GC, and the peak exo power reached 1.4% GC after the peak of biological ankle plantarflexion torque and power, respectively. In [33], an example of a proportional myoelectric controller without optimization is presented. Peak exo torque and power occurred 9.8% GC and 2.0% GC after peak biological torque and power, respectively. In [4], a passive ankle exo is featured. This time, the peak exo torque and power both preceded their biological counterparts by around 1.2% GC. In all three examples, the exo torque trajectories resembled the scaled-down biological torque trajectories. Both soleus muscle activity and metabolic cost were reduced in all cases. Based on these results, we estimate that a well-synchronized biomimetically shaped torque trajectory should likely reach its peak between −1% GC and 10% GC after the biological torque peak. The timing constraints for the power appear much stricter at −1% GC to 2% GC after the biological power peak.

In our study, the peak exo torque occurred on average 0.96 ± 0.07% (mean ± standard error of mean) GC before peak biological torque. Peak exo power occurred 2.41 ± 0.04% (mean ± s.e.m.) GC after peak biological power. These timings are very similar to those in [2] and highly repeatable across different assistance levels and users.

In a rehabilitation setting, we argue that this high degree of repeatability is especially important, since the device should behave in a safe and robust way at all times. The assistance synchrony achieved by AN-EXTRA-Push even in spite of intersubject repeatability bodes well for use with patient populations, where a lot of intrasubject variability in gait patterns is expected. This stands in contrast to HILO-optimized controllers, usually researched in the context of human augmentation, which rely on regularity of gait.

### Study Limitations and Generalizability of Results

We acknowledge that our study has limitations. The main limiting factor in subject recruitment was the fact that the AN-EXTRA-Push prototype used in this study could only be used by people with EU shoe sizes 42 to 46. This resulted in a cohort of only males with similar anthropometric characteristics, not representative of the general population. For future studies and potential integration of the device in clinical protocols, a new version of the device will be produced, featuring a large range of shoe size options and shank attachment point locations. With it, we will be able to adjust for much more anthropometric variability, enabling us to recruit a more diverse and gender-balanced study cohort.

Participants were forced to diverge from their preferred gait pattern due to how the treadmill was constructed. The belts of the split-belt treadmill were spaced 24 mm apart, which imposed a wider than preferred gait pattern onto the participants. The non-instrumented belt was only 81 cm long, which was sufficient to enable walking with a typical preferred stride length [41], but restricted whole-body anteroposterior movement. Additionally, all limitations concerning the ecological validity of treadmill walking for the purpose of improving overground gait [42] are inherent to our device.

While wearing the AN-EXTRA-Push orthosis, a reflective marker could not be placed at the location on the shank that is prescribed by the Plug-In Gait model. The marker was instead placed on the orthosis, and the model was modified accordingly.

In spite of the homogeneity of this study cohort, we observed large intersubject variability in measured gait patterns, including during the ZeroTorque condition. AN-EXTRA-Push assistance during active conditions affected all participants in substantially similar ways, with minor deviations from the results reported for the average gait cycles shown in Figure 3, Figure 4, Figure 5 and Figure 6. Crucially, the exo assistance remained synchronous with the participants’ own efforts across their different gait patterns, as explicitly shown in Figure 7 and Figure 8.

This study only included able-bodied individuals, so any clinical translation is so far speculative. While the effects of AN-EXTRA-Push were consistent with our expectations and comparable to state-of-the-art ankle exos used in rehabilitation [29,30], clinical studies on patient groups with specific impairments are the necessary next step.

## 5. Conclusions

In this study, AN-EXTRA-Push performed its intended function of providing adjustable levels of plantarflexion torque assistance. In our able-bodied subjects, this resulted in reduced user effort that was reflected in diminished biological power at the joints and decreased ankle plantar flexor muscle activity, without deleterious effects on the overall gait kinematics or dynamics.

The torque and power-generating capacity of AN-EXTRA-Push was comparable to state-of-the-art active ankle exos. The assistive torque trajectories were similar both in shape and magnitude. This was achieved using biomimetic mechanical design rather than a complex control algorithm. The inherent synchrony of the exo assistance with the user’s own efforts, combined with the low complexity of the device, represents a step towards an effective, reliable ankle exo for use in gait rehabilitation in clinical practice.

## Figures and Tables

**Figure 1 biomimetics-10-00635-f001:**
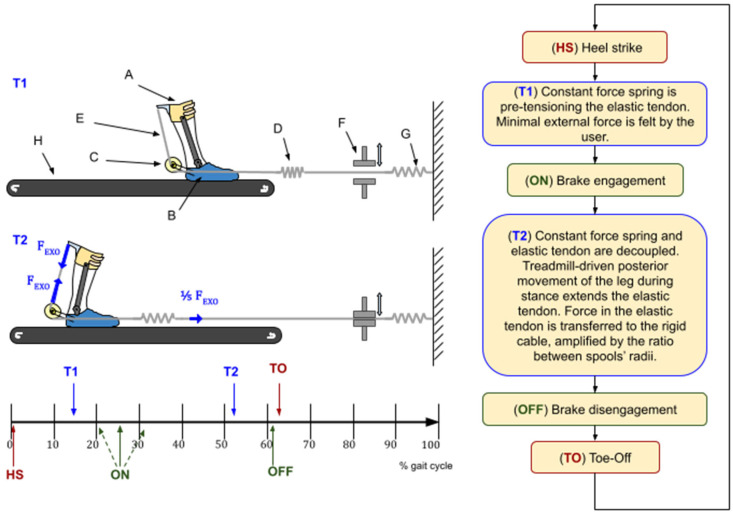
AN-EXTRA-Push structure (**top left**) and normal operating procedure. A: shank part of the orthosis; B: foot part of the orthosis; C: three rigidly affixed coaxial spools; D: elastic tendon; E: rigid cable; F: brake; G: constant force spring; and H: treadmill (not part of the exo). Descriptive moments T1 and T2 are depicted on the left. Key moments are placed on the timeline of a gait cycle (**bottom left**) and explained in a flowchart (**right**).

**Figure 2 biomimetics-10-00635-f002:**
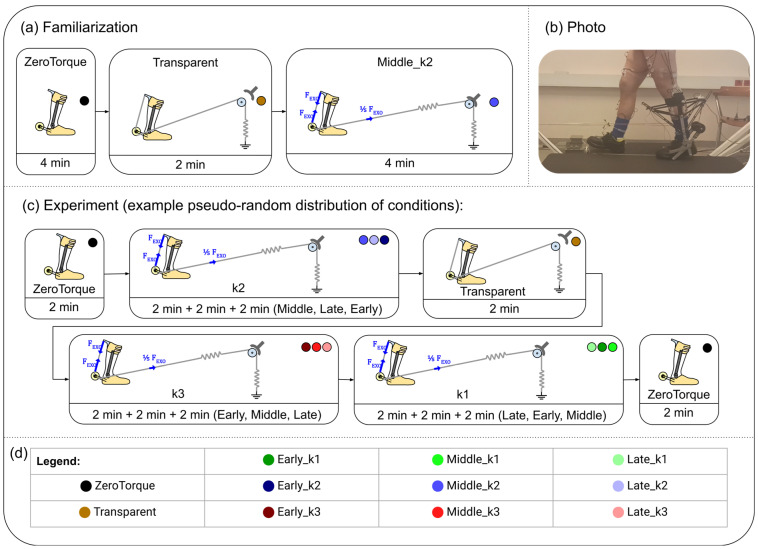
(**a**) Familiarization protocol. (**b**) Photo taken during the experiment. A video clip of the experiment is available in Appendix A. (**c**) Example of the experimental protocol. (**d**) The legend shows the color code for each experimental condition. The same color coding is applied across all figures in this paper that depict results.

**Figure 3 biomimetics-10-00635-f003:**
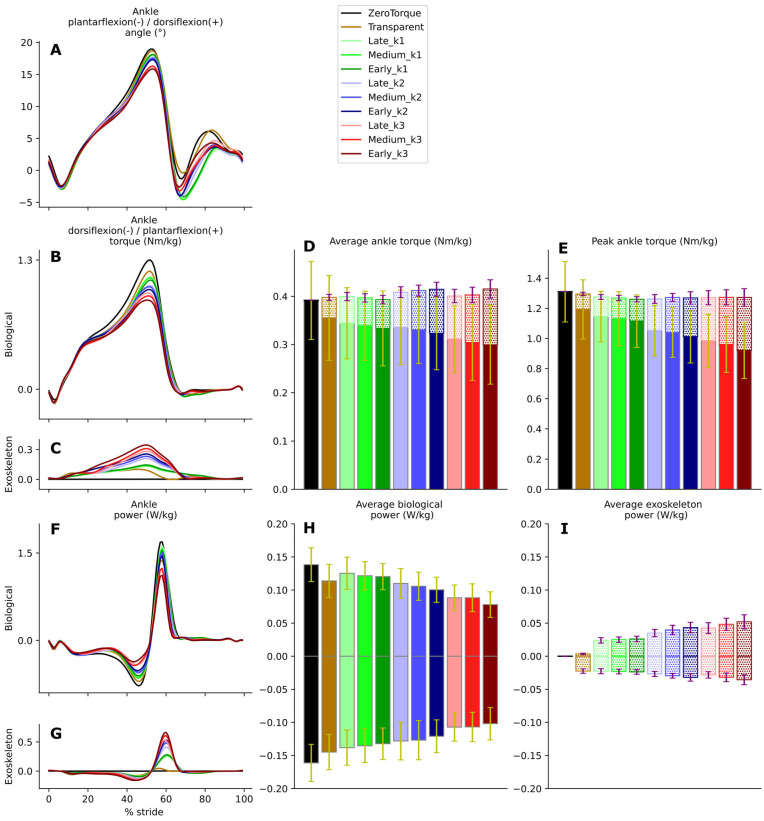
Ankle biomechanics. Timeseries: Angle (**A**), biological torque (**B**), exo torque (**C**), biological power (**F**), and exo power (**G**). Stacked bar charts: average biological (full color, yellow error bars) and exo (hatched, purple error bars) torque over stride (**D**); peak biological (full color, yellow error bars) and exo (hatched, purple error bars) torque (**E**); average positive and negative biological power (summed separately, yellow error bars) (**H**); and average positive and negative exo power (summed separately, purple error bars) (**I**).

**Figure 4 biomimetics-10-00635-f004:**
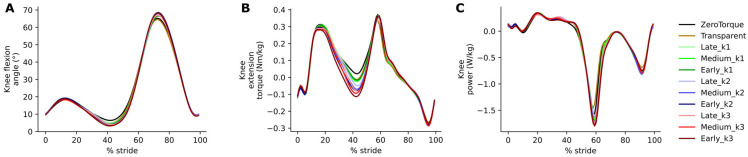
Knee angle (**A**), torque (**B**), and power (**C**).

**Figure 5 biomimetics-10-00635-f005:**
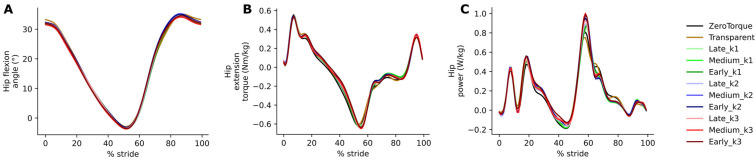
Hip angle (**A**), torque (**B**), and power (**C**).

**Figure 6 biomimetics-10-00635-f006:**
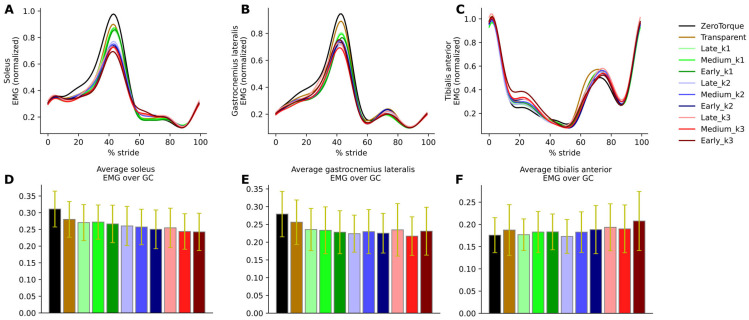
EMG results: Soleus (**A**,**D**), gastrocnemius lateralis (**B**,**E**), and tibialis anterior (**C**,**F**) normalized to peak EMG value during ZeroTorque. The timeseries shown in the upper row of panels were integrated and averaged over GC to produce bar charts in the lower row of panels. Yellow error bars show ± one standard deviation.

**Figure 7 biomimetics-10-00635-f007:**
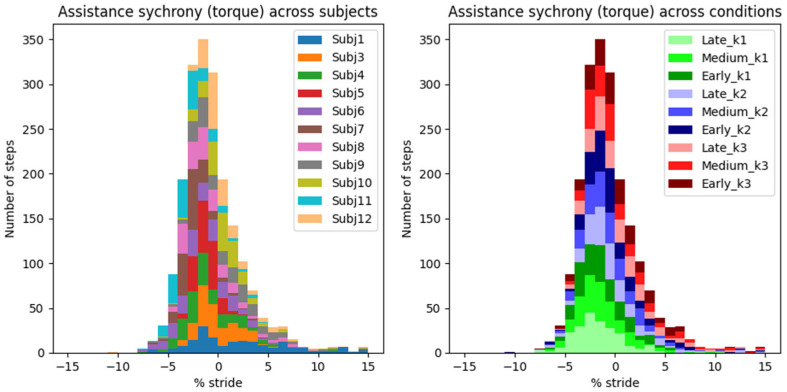
Assistance synchrony computed as the time lag between the peak exo’s assistive and the user’s biological ankle plantarflexion torques during push-off. Data are grouped by user in the left panel and by experimental condition in the right panel.

**Figure 8 biomimetics-10-00635-f008:**
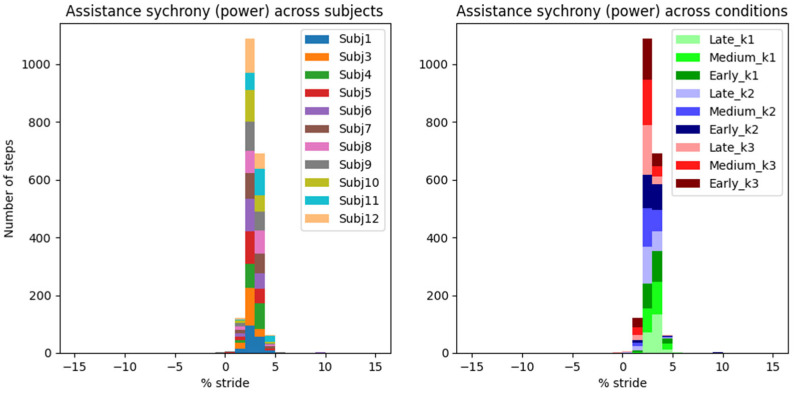
Assistance synchrony computed as the time lag between the peak exo’s assistive and the user’s biological push-off power. Data are grouped by user in the left panel and by experimental condition in the right panel.

## Data Availability

The raw data supporting the conclusions of this article will be made available by the authors on request.

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
