# Peer review of "A Biomimetic Treadmill-Driven Ankle Exoskeleton: A Study in Able-Bodied Individuals"

_biomimetics, 2025, doi:10.3390/biomimetics10090635_

Round 1
Reviewer 1 Report
Comments and Suggestions for Authors
This study introduces and evaluates AN-EXTRA-Push, a biomimetic treadmill-driven ankle exoskeleton designed to provide plantarflexion push-off assistance during gait. The device integrates features of both active and passive exoskeletons—combining power generation with inherent synchrony—using a mechanically simple brake–spring system. Twelve able-bodied individuals participated (11 valid datasets), walking under 11 experimental conditions that varied brake engagement timing and tendon stiffness. Biomechanical, kinetic, and EMG data were collected using motion capture, force sensors, and EMG electrodes. Results showed that the exoskeleton provided adjustable levels of plantarflexion torque (10–28% of peak ankle torque) and reduced biological torque and soleus activation in proportion to assistance. Knee kinematics were slightly altered, but hip biomechanics and contralateral leg parameters were largely unaffected. The newly proposed assistance synchrony metric demonstrated high repeatability of torque/power timing between exoskeleton and user. The authors conclude that AN-EXTRA-Push achieves assistance levels comparable to active devices while maintaining simplicity and synchrony, supporting its potential for rehabilitation applications.
- The concept of combining passive synchrony with active power input is interesting. However, the introduction should more clearly highlight how AN-EXTRA-Push improves over existing passive/active ankle exoskeletons in rehabilitation contexts.
- More details on device limitations (e.g., shoe size restrictions, treadmill setup constraints) are buried in the discussion. These should be stated earlier to frame interpretation of results.
- The proposed metric is valuable, but more justification is needed on why time-lag between peaks was chosen over cross-correlation or continuous similarity metrics.
- The study tested only young healthy males. The lack of diversity in sex, age, and clinical participants is a major limitation that should be emphasized.
- While torque reduction and soleus unloading are consistent with expectations, the clinical translation (e.g., for stroke survivors) is speculative. Please discuss potential risks (e.g., maladaptation, over-reliance).
- Figures are data-rich but may be overwhelming. Consider simplifying legends and providing clearer summaries of key findings.
- The discussion does a good job referencing prior work, but direct comparisons (e.g., absolute vs. relative torque support, device weight, complexity) could be presented in a concise table.
- How robust is AN-EXTRA-Push performance across users with more variable gait patterns (e.g., stroke patients with asymmetry)?
- Could the synchrony metric be extended to evaluate the entire torque trajectory instead of just peak values?
- Was metabolic cost measured, or do you plan to test it in future studies?
- How easily can AN-EXTRA-Push be adjusted for patients of different sizes, weights, or gait speeds?
- What are the safety considerations (e.g., unexpected brake malfunction) for clinical application?
- Quality of English: The English is clear and professional, with appropriate scientific terminology for biomechanics, rehabilitation robotics, and human–machine interaction. The manuscript is readable and well-structured, though some sentences are long and could be simplified for better flow. A few minor grammatical and stylistic issues (such as article use, run-on sentences, or redundant phrasing) should be corrected. Overall, the English quality is good and does not obscure the meaning, but language editing would enhance conciseness and readability.
Reviewer 2 Report
Comments and Suggestions for Authors
Recommendations:
- The authors must consider shortening the title or making it more clinically oriented.
- Regarding the Abstract: the authors can Add a final sentence with practical implications (e.g., how the device could be used in rehabilitation trials or clinical practice).
- Introduction
- It is recommended that the authors strength the gap statement: currently, they note the lack of clinical adoption, but it would help to emphasize why—cost, complexity, weight, control requirements.
- The authors must mention briefly how the biomimetic + treadmill-driven hybrid design specifically addresses these barriers (clinical feasibility, simplicity).
- Materials and methods
- The authors must provide more justification for 0.8 m/s treadmill speed in terms of patient comparability.
- Results
- The figures are ok, but:
- It is recommended that the authors use consistent color schemes for biological vs. exoskeleton contributions across all plots (yellow vs. purple is good but should be clearly labeled in each figure).
- Some bar plots have dense labels—adding supplementary enlarged versions may help.
- Discussion
- The authors must provide a short outlook: how might the device be scaled for patient populations, adjustable for anthropometric variability, or integrated into rehabilitation protocols?
- Limitations
- There are already covered, but the authors might expand on:
- Device constraints: treadmill-bound, limited shoe sizes.
- Ecological validity: treadmill walking differs from overground walking.
- Language and style
- It is recommended to ensure consistency of terms: exo vs exoskeleton: unify.
- Small grammar refinements:
- It is recommended to replace “we’ve seen numerous researches” with “numerous studies have reported”.
- It is recommended to replace “does not allow for much anteroposterior movement on the treadmill” with “restricted anteroposterior movement”.
Round 2
Reviewer 1 Report
Comments and Suggestions for Authors
Thank you for addressing the comments.
Reviewer 2 Report
Comments and Suggestions for Authors
following the changes made, the paper can be published